# The Impact of a Dementia-Friendly Exercise Class on People Living with Dementia: A Mixed-Methods Study

**DOI:** 10.3390/ijerph17124562

**Published:** 2020-06-24

**Authors:** Annabelle Long, Claudio Di Lorito, Pip Logan, Vicky Booth, Louise Howe, Vicky Hood-Moore, Veronika van der Wardt

**Affiliations:** 1Division of Rehabilitation, Ageing and Wellbeing, School of Medicine, University of Nottingham, Nottingham NG7 2UH, UK; claudio.dilorito@nottingham.ac.uk (C.D.L.); pip.logan@nottingham.ac.uk (P.L.); Vicky.Booth@nottingham.ac.uk (V.B.); louise.howe@nottingham.ac.uk (L.H.); Victoria.hood-moore@nottingham.ac.uk (V.H.-M.); v.vanderwardt@uni-marburg.de (V.v.d.W.); 2Department of General Medicine, Preventative and Rehabilitative Medicine, Philipps-Universität Marburg, Karl-von-Frisch-Strasse 4, 35043 Marburg, Germany

**Keywords:** dementia, exercise, mixed methods

## Abstract

Exercise has multiple benefits for people living with dementia. A programme of group exercise classes for people with dementia and their family carers has been established in a University sports centre. This study aims to explore the impact of this programme on participants with dementia and their carers. A mixed-methods design including a prospective, repeated measures cohort study followed by focus groups was employed. Physiological and cognitive outcome measures were repeated at baseline and three months in a cohort of people with dementia attending a group exercise class. Focus groups on the participants’ experiences and their perceptions of the impact of the exercise class on their lives were then conducted. The results were analysed and mapped on a model, to illustrate the components that most likely promote participation. Sixteen participants (n = 8 with dementia, and *n* = 8 carers) were recruited, and completed both baseline and follow up assessments. Positive mean differences were found in physical activity (4.44), loneliness (1.75), mood (1.33) and cognition (1.13). Ten participants were included in the focus groups, which found that accessibility of the exercise venue, opportunities for socialisation and staff who were experienced working with people living with dementia were key to participants reporting benefits. The four key themes from the focus group data were synthesised to produce a model outlining the components that might generate a positive impact of the exercise classes and promote participation. Exercise classes for people with dementia can be delivered with success in novel environments such as University sports centres. There is some indication of improvement over a short period of time. The model derived from this study will inform strategies to promote attendance at dementia-friendly exercise classes.

## 1. Introduction

There are over 850,000 people living with dementia in the UK [1]. Exercise has shown multiple benefits for people living with dementia, including improvements in mood and mobility in activities of daily living, as well as reduction of behavioural problems and risk of falls [2,3,4]. However, the effect of any intervention depends on adherence. A systematic review by Vseteckova et al. found adherence rates varied widely from 25.5% to 84%. They concluded that supported physical activity led by knowledgeable staff was crucial for adherence in people living with dementia [5]. Group participation can also support engagement and adherence in exercising [6,7] as well as reducing social isolation and loneliness [8]. However, there is little provision to support this population to maintain their physical health through exercise.

Many exercise interventions provided by community services are not designed nor tailored for people living with dementia [9]. In 2017, from ongoing research of an intervention to support independence, activity and stability in people with dementia [10], the University of Nottingham established a group exercise class in its sports centre for people with dementia and their family carers. Attendance has increased from six participants in September 2017 to up to 30 participants in October 2018.

Informal feedback from participants indicates that they value the chance to be physically active, the challenge of the exercises and the camaraderie with the other participants [11]. However, the impact of the exercise group on the participants and their family carers has not been formally investigated.

Therefore, this mixed-methods study aimed to explore the impact of an established dementia-friendly exercise class on the participants with dementia and their carers. The objectives were to:Assess the impact of an exercise class on mobility, muscle strength, physical activity levels, independence, loneliness, mood, and quality of life for people with dementia and their carers.Investigate the perception of the impact of an exercise group from the perspective of people with dementia and their carers.Map the key components related to impact and participation of the exercise classes on people with dementia and their carers.

## 2. Materials and Methods

### 2.1. Design

A mixed-methods design was used, including a prospective, repeated measure cohort study followed by focus groups. Initially, outcome measurements were collected across two time points (March 2019 and June 2019) on a cohort of people with dementia and their carers who attend a dementia-friendly exercise class (as described above). Participants of the cohort study were then invited to focus groups.

Ethical approval was obtained from the Faculty of Medicine and Health Sciences Research Ethics committee of the University of Nottingham (Ref No. 203-1902).

### 2.2. Participants

The inclusion criteria for the study were: a person with dementia (any stage) and his/her carer; both attending the dementia-friendly exercise class; both able to give informed consent. All those meeting these criteria were considered eligible and invited to take part in the study.

### 2.3. Setting

The exercise class takes place once a week in a studio of the David Ross Sports Village (DSRV), University of Nottingham (UoN). The DSRV is purpose built to support a range of sports and exercise for university students, staff, and members of the public. It is easy to access (e.g., ground floor, level access, next to disabled toilets), has free parking for two hours, and has an onsite cafe.

The class is led by a registered physiotherapist, with trained volunteer helpers. It was established in 2017 and provides an on-going, rolling programme. It is offered by the UoN and does not require referral or pre-booking for participants to take part (e.g., Referral through General Practice). To ensure equal opportunities of access to participants, the class is free of charge and does not require participants to have a gym membership. The exercise class is grounded in the framework of positive support developed by Clare [12] for rehabilitation of people with dementia. The framework emphasises the importance of creating rehabilitation opportunities that provide positive experiences and include the support of family carers.

Each class is an hour long and follows a consistent structure. The class starts with a five minute instructor-led warm-up consisting of cardiovascular work (e.g., marching and running on the spot) and movement of major muscle groups. The class then becomes a small group (two to six participants) six-station circuit, completed for one minute and repeated twice. Each station features the same exercise each week and can be completed at different levels, depending on individual physical capability, and personal progress. Exercises at the stations include squats, step ups, bicep curls with weights, push-ups, boxing, and shuttle runs. An example of the stations was described in Appendix A. Each station is completed for one minute before moving to the next, until a full circuit is completed. A timed five minute break encourages participants to have a glass of water, before the circuit is repeated. Carers are encouraged to support the person with dementia if they wish to.

The class finishes with 10-min instructor-led balance exercises, including static and dynamic postures (e.g., tandem walking, backwards walking and single leg stands). The facilities of the DSRV are available for use following the class (e.g., changing rooms, café), providing additional opportunities for the attendees. The class content was identified from available evidence and clinical experience [13] and adapted for people with dementia by several measures (e.g., increased levels of supervision and support, consistency of structure, drop-in on-going programme and accessible location).

### 2.4. Procedures

Potential participants were informed about the study by one of the volunteers or the lead physiotherapist during the weekly class. Any exercise class attendees who were interested in taking part were introduced to a member of the research team, who explained the study and provided participant information sheets.

#### 2.4.1. Repeated Measures Cohort

Participants provided informed consent and baseline measures were collected during a three-week data collection phase in March 2019 (objective 1).

Outcome measures included:(a)Mobility: Berg Balance Scale [14] and the Timed Up and Go (TUG) [15](b)Muscle Strength: hand held dynamometer [16](c)Cognition: Hopkins Verbal Learning Test (HVLT) [17](d)Physical activity levels: LASA Physical Activity Questionnaire (LAPAQ) [18](e)Independence: Nottingham Extended ADL Scale (NEADL) [19](f)Loneliness: UCLA Loneliness Scale [20](g)Mood: Hospital Anxiety and Depression Scale (HADS) [21](h)Quality of Life: Dementia Quality of Life Scale (DemQoL) [22]

The mobility (a), strength (b), cognition (c) and quality of life (h) assessments were completed with the participants before, during or after the exercise class (depending on their preference) by one of the researchers (AL, VB, LH). Participants were then given the other questionnaires (d–g) to be completed at home and returned to the research team at the next session. Participants were encouraged to complete them with their carer, where indicated.

All measures were repeated three-months later (June 2019) by the same researchers (AL, VB, LH) following the same procedure as baseline.

#### 2.4.2. Focus Groups

All participants were invited to take part in one of two focus groups facilitated by the research team (objective 2). An interview schedule was used to guide the discussion of their experiences of the exercise class, as well as their perceptions of the impact of the exercise class on their lives (Appendix B). The schedule was used as a template, but a flexible approach was adopted, to enable the discussion of any (relevant) new topic emerging during the session [23].

Two focus group sessions (morning or afternoon) were offered to participants to take account of peoples’ differing needs and capabilities during the day. The sessions took place at a location within the University of Nottingham with free accessible parking. Each focus group had two experienced researchers present: one to facilitate the session (AL); a second to take field notes to inform and assist with transcription (LH). Both focus groups were recorded on a digital audio recorder.

### 2.5. Data Analysis

#### 2.5.1. Repeated Measures Cohort

All data were entered into SPSS 24.0 [24]. Descriptive statistics were used to describe the sample. Mean differences were used to describe changes across outcomes over the two time points.

#### 2.5.2. Focus Groups

Focus group data were transcribed verbatim, checked and fully anonymised by a member of the research team (AL) using the audio files and field notes for accuracy. Audio files and transcription files were uploaded onto NVivo 12 [25]. Each transcription was coded by one researcher (AL) using a thematic, inductive yet flexible coding process [26]. If during the coding process, novel themes emerged from the transcripts, new codes were generated. All codes were mapped and subject to expansion, restriction or modification through use of concept maps within Nvivo 12 [25], until the research team (A.L., C.D.L., V.v.d.W.) agreed on a final list of codes. A code book (Table A1 in Appendix C) was then developed. The data were then extracted and coded by one researcher (AL)

Data was then used to develop an ad hoc model which identifies the conditions that generate a positive impact of the exercise classes on the participants and that promote their continuing participation. Based on this objective, two researchers (A.L. and C.D.L.) (independently of each other) extracted from the transcripts of the focus groups the components that the participants reported as contributing to having an impact on them. It was assumed that these would lead to their continuing participation over time. The two researchers convened and synthesised the identified variables by umbrella themes (e.g., the variables accessible toilets and accessible class were synthesised into the umbrella theme “accessibility of venue”). If during the process, the two researchers disagreed on coding, a third researcher was involved (V.v.d.W.), until consensus was reached.

The three researchers then mapped the identified umbrella themes into a diagram. This required several iterations, until a final version was agreed upon by the research team.

## 3. Results

### 3.1. Repeated Measures Cohort

Sixteen participants (*n* = 8 living with dementia, *n* = 8 carers) were recruited and consented to take part in the physiological and cognitive outcome assessments. All participants were recruited in dyads. There was no attrition of participants, as all of them completed baseline and the follow-up assessments (*n* = 16, 100%). Of the eight outcomes included within the repeated assessments, four were not completed at baseline (3%) and three at follow up (2%). One DEMQoL at baseline, and one HADS at follow up were not completed due to the partcipants distress. Three UCLA loneliness scales at baseline and two UCLA loneliness scales at follow up were not completed due to both carer and participant difficulties with the questions.

The participants’ demographic information is reported in Table 1. Fifty-six percent (*n* = 9) of all participants were female. Of the participants living with dementia, six (75%) were male and five (62.5%) had a diagnosis of Alzheimer’s disease.

Following a three-month attendance at a dementia-friendly exercise class, there were improvements in physical activity levels, loneliness, cognition, anxiety and depression, for all participants. Strength and activities of daily living remained largely unchanged whilst mobility, balance and quality of life slightly deteriorated (Table 2). However, because of the small sample size, descriptive statistics were used rather than inferential statistics.

Over the three-month period, physical activity levels improved by 4.44 met hrs per week (45.42–49.86) which, although not statistically significant, does equate to a 10% increase in activity over a three–month period. The participants ability to recall improved by a score of 1.13 (15.63–16.75), but recognition remained static, with an improvement of 0.06 (7.88–7.94). Both loneliness (1.75 points, 44.17–42.42) and mood (1.33 points, 13.00–11.67) decreased.

Activities of daily living showed minimal changes of >1 point, with independence scores decreasing by 0.75 (14.31–13.56). Grip strength in the right hand decreased by 0.29kg (24.44–24.15), and in the left it improved by 0.89kg (22.81–23.69).

The participant’s quality of life showed a small deterioration as scores decreased by 1 point (93.00–92.00). However, that reflects an overall decrease in quality of life of less than one percent (0.86%). Balance also deteriorated slightly over the three-month period, with a reduction of 1.25 points (50.38–49.13).

Initial descriptive statistics indicated that mobility had deteriorated over time with an increase of 1.29 s (9.72–11.02). However, there was a significant outlier in the follow-up data (follow-up TUG 40.72 s for a participant with Lewy Body Dementia). The TUG was reanalysed with the outlier removed, subsequently showing a positive trend towards improvement with a decrease of 0.48 s (9.52–9.04).

### 3.2. Focus Group

#### Description of Sample

Ten participants were included in the two focus groups (*n* = 5 people living with dementia, *n* = 5 carers). All participants attended as dyads. Demographic information for the focus group participants are reported in Table 3. The remaining six participants from the repeated measures cohort did not want to attend this phase of the study due to other commitments. The first group (FG1) consisted of four participants (*n* = 2 people living with dementia, *n* = 2 carers) and the second (FG2) of six participants (*n* = 3 people living with dementia, *n* = 3 carers).

### 3.3. Themes

Data analysis identified four key themes: benefits to the person living with dementia, benefits to the carer, environmental characteristics, and exercise class characteristics.

#### 3.3.1. Benefits to Person with Dementia

All participants reported that the classes had helped them maintain their current abilities and prevented deterioration.


*Well I think it’s kept my mum continue to be mobile……..she can still go upstairs. So I don’t have to do anything for her. She gets dressed and undressed. So she’s still got muscle strength. So perhaps that would have deteriorated if she hadn’t been coming to the classes.*
(FG1 P1)

The participants also felt that attending the class was increasing their levels of activity by either encouraging them to participate in other activities or enabling them to continue with or return to previous activities.


*It’s almost like building up a bank of things…..You pick your activity just like you or I would do it. You wake up one day, you feel like doing something or you don’t. And it’s just to have the option of different things I think.*
(FG2 P3)

Some participants also felt that the classes increased their confidence and helped them be more willing to participate in other activities, even if they were not advertised as being dementia friendly.


*We try everything……I say….we’re going to go here, if you don’t like it we won’t go again, simple as that. That’s all you can do, but you’ve just got to give it a go, because what suits one doesn’t suit another does it?*
(FG2 P3)

#### 3.3.2. Benefits to Carer

Although carers did mention the impact of doing the exercise for themselves, the main impact that carers commented on was on the supportive community that the class had provided and the sharing of useful knowledge between carers.


*The things I’ve picked up have been so good. And the people we’ve got to know, you get really friendly with people in a short time, because you’re in a mess together, and you pick brains. And one of the guys told us about, he just said are you aware that you can get money back if you’ve altered powers of attorney?*
(FG2 P5)

Many carers also commented on the changes that occurred in their support networks following the dementia diagnosis.


*Look here, we’ve met these lovely people I never want to be without.*
(FG2 P1)

They also commented on how burdensome it is to be a carer, dealing with the lack of sleep and fatigue and also dealing with the diagnosis itself. However, they felt the group provided support as everyone was on a similar dementia journey.


*Well the support, the people’s experience and their knowledge….the little bits of information that we got, I felt quite desperate to start with, what are we going to do? I’m always picking up on what people say.*
(FG2 P5)

Other participants valued just being able to get out of the house and having social interaction with other people.


*Well getting out the house for a start. Sort of, I don’t know, the older you get you can’t be bothered to go out*
(FG1 P1)


*Well it’s mixing with other people basically……. and we have a lot of fun.*
(FG2 P6)

#### 3.3.3. Environmental Characteristics

Both groups felt that the environment the class was held in had an impact on how they felt about the class and contributed towards their continuing attendance.

The availability and accessibility of parking and toilet facilities as well as the opportunity to sit and have coffee with other participants after the class were seen as key elements for coming to the class.


*I was a bit worried for quite a while about the parking, because I couldn’t make out where we would park. Anyway…….there is parking…….*
(FG1 P3)


*There’s loads of toilets. And there’s disabled toilets. So I don’t think they’re lacking in anything are they really?*
(FG1 P1)


*So to be able to extend it and know that John will have a lovely chat with his friends as well that just makes the whole experience last longer in my view.*
(FG2 P3)

It was acknowledged that although some people liked to attend on their own, giving their carer time to go and do something else, some people liked to attend with a partner, friend, carer or family member. Having volunteers who were knowledgeable about dementia were important factors in giving people the confidence to attend.


*They’ve just got so much confidence. They seem to know what they’re doing. And he enjoys it far more when he’s with somebody else than me.*
(FG2 P3)


*I wouldn’t think of taking my mum to any exercise class unless it said it was specifically for dementia………you’d know that the people doing it would know about dementia and all what happens to people when they’ve got dementia and the different kinds and…. they’d know what to do*
(FG1 P1)

#### 3.3.4. Exercise Class Characteristics

The class was advertised as being specifically designed for people for dementia, and this was important to most of the participants who felt that they would not have been comfortable attending an exercise class designed for older people in general.


*Well it said it was dementia friendly so I just assumed it would be……I hadn’t really wanted to go to things that people said oh it might have been with dementia and their partners, it just didn’t seem to appeal to me much. So I wasn’t very motivated to go to any of that, but I thought well an exercise class, anyone can do that. I’ll enjoy doing that as well.*
(FG1 P3)


*I wouldn’t think of taking my mum to any exercise class unless it said it was specifically for dementia.*
(FG1 P1)

Participants also felt that holding the class once a week was very beneficial as they attended other activities that took place once a fortnight or monthly and felt that these were not frequent enough.


*Forget me Nots does…..walking football, so we can do that once a month. But once a month, it’s not enough *
(FG2 P3)


*And it’s nice the once a week as well, because a lot of things are only once every three weeks or four weeks. To have this nearly every week, and it’s just awesome. *
(FG2 P3)

Another relevant intervention characteristic that participants appreciated was the flexibility to accommodate the needs of participants at different levels of capability.


*It’s just the whole package. You can do as much or as little as you like. You can see that it’s got so much mileage in the class, because even if either you deteriorate or you’re not feeling great, you can just do whatever you want to do. It’s adaptable. And you know what, even if you do very little it’s still a social thing. *
(FG2 P1)

### 3.4. Exercise Class Model

The four key themes from the focus group data were synthesised and the results are shown as a model (Figure 1), outlining the components that might generate a positive impact of the exercise classes and promote participation.

The key ingredients which participants said promoted their participation in the group class were: accessible parking and toilets, a coffee area to socialise after the class, and the support of volunteers. Several exercise-class characteristics were highlighted in the focus groups as being required for effective delivery of the class. These were: the class was designed for people with dementia, carers could attend, instructors and volunteers’ being knowledgeable about dementia and had experience of delivering services to people with dementia, and/or caring for someone with the condition.

Because of all these characteristics being present, a cycle was set in motion. The participants with dementia and their carers felt motivated to take part in the class; this produced a positive impact on them, which encouraged attendance to the class. Their attendance ensured, in turn, that the key ingredients required to encourage participants to take part were reinforced (e.g., instructors’ skills to encourage involvement in the class).

## 4. Discussion

This study showed that exercise classes specifically designed for people with dementia can be delivered in a novel environment, such as a University sports centre. In addition, some cognitive and physiological abilities improved as a result of the attendance to the dementia-friendly exercise class over a three-month period. Importantly, most measures did not decline over time, which, in people with dementia, is to be considered a positive outcome. This study also showed that people with dementia and their carers enjoyed attending an exercise class and that key components were identified which maintained their participation over a notable time period.

Critical for joining the class and achieving a positive impact were several environmental and exercise-class prerequisites. These included the knowledge that the class had been developed for people with dementia and their family carers, and the fact that it was facilitated by professionals and volunteers with knowledge of and previous experience with dementia. The participants felt motivated and comfortable about joining the group, as they knew that all other participants would have a shared experience with dementia. This shared experience of dementia for the participants and volunteers is an important, mutually beneficial relationship already identified in the literature [27]. Other crucial practical aspects to promote attendance were easy access (parking close by) and the opportunity to socialise in a café. This reflects the results of a study exploring barriers to leisure participation in people with dementia, which found good transport opportunities to be a central feature for engagement in activities [28]. The café offered a chance of communication with people on a shared journey, opportunities for carers’ respite and access to readily available peer-support. These aspects have been shown to serve as motivators for people with dementia and their family carers [28,29] and align with the principles of a dementia friendly community [30].

Considering the multiple benefits of exercise [2,3,4], it is important to enable people with dementia to participate in exercise groups. Group exercise will not appeal to everyone [31], and some people with dementia might prefer groups designed for the general population, but the steadily increasing numbers of these exercise classes [32] show that there is an interest in a physiotherapist-led exercise group, if the contextual conditions are met. While it is not possible to design a group exercise class that appeals or is fully accessible to everyone, the model developed on the basis of the findings of this study outlines a set of conditions that may support the motivation of people with dementia and their family carers to attend.

This study confirmed some of the findings reported in the existing literature and contrasted with others. Despite using different outcome measures, Barnes et al. [33] also found a small, non-significant improvement in physical and cognitive ability following an 18-week group exercise programme. However, unlike Barnes [33], this study identified a deterioration in mobility and quality of life measures. Other group exercise studies in dementia populations have also reported negative effects in some outcomes [34]. These contrasting findings could be due to the progressive nature of dementia, indeed, similar group exercise studies that have reported the maintenance of ability as a positive result [4]. Additionally, there is some doubt that the population was representative of a dementia population. Telenius [35] reported a mean Berg Balance Score of 38 ± 13.7 in their sample of residents with mild-to-moderate dementia, which is significantly lower and therefore less physically able than this sample.

This study had certain strengths and limitations. It would have been preferable to recruit new members to the class, as opposed to people who had attended for some time. This might have indicated a larger impact on the outcome measures, particularly as the length of time the participants had been attending the class was not documented [30]. However, most of the outcomes improved, which in people with dementia is seen to be a substantial benefit as they are usually experiencing a decline over time [4]. Another limitation was that only people who had already chosen to participate in the exercise class were included. As a result, we do not know the views of others who had not joined or had only attended once. A small sample size is also a limitation to the study. Outlying results in a small group can lead to skewed results, as seen with the mobility scores. The design of the study (non-randomised, cohort with no control group) is a further limitation that might influence the generalisability of findings, particularly as certain demographic information was not collected to fully describe the sample (e.g., age). However, that was not the intention of the study, which aimed to explore and map the impact of an existing exercise class on people with dementia and their carers. Future research should be conducted including control groups to further validate these findings.

The model that we developed also has limitations. Because it is based on study results, it does not provide an exhaustive list of the components required to ensure impact and promote engagement and is highly time and place bound. However, the model is only to be intended as a preliminary framework, which will require several iterations and a formal validation study before it can be generalised to other settings. In particular, there is a need to test its validity in a wider and more diverse group of people with dementia and their family carers (e.g., ethnicity, diagnosis of dementia), and in different settings (e.g., inner city urban, rural communities).

Despite these limitations, there are several strengths to this study, including the utilisation of a mixed-methods design to fully investigate the phenomenon of interest [36]. This is particularly relevant when exploring complex interventions in heterogeneous populations [37]. Whilst the convenient sample of people already involved in an exercise class has limitations [38], it ensured that the participants were engaged in exercise and provided additional data for qualitative modelling.

Another strength of the study is that it clearly identified key components that others who are wishing to implement exercise classes for people with dementia should consider for inclusion: being able to attend with the family, carer, friend or alone, being able to undertake different exercises at an individual pace supervised by a trained person who is knowledgeable in dementia care and exercise. This type of person-centred exercise has been recommended by others [39,40] and whilst it was not delivered by a multidisciplinary rehabilitation team as suggested by Ravn [40], the results of this study indicated that the participants created their own multidisciplinary programme by combining the exercise class with other activities.

## 5. Conclusions

Novel environments, such as university sports centres, appear to be suitable environments to deliver exercise to underserved populations, such as people with dementia. If key environmental characteristics (i.e., class designed for people with dementia and carers, facilitated by professionals and volunteers with experience with dementia, easy access, availability of parking and a social area) could be replicated in other venues, participation in exercise classes is likely to produce a positive impact both for the person with dementia (e.g., maintaining current abilities and preventing deterioration, increasing confidence and chances to participate in other activities) and the carer (e.g., supportive community, sharing of useful knowledge).

## Figures and Tables

**Figure 1 ijerph-17-04562-f001:**
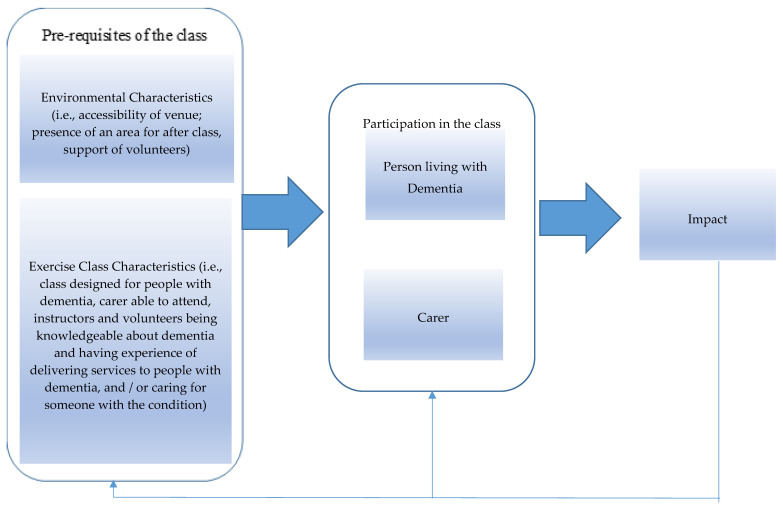
A visual model synthesising the components that generate a positive impact of the exercise classes on the participants and that promote their continuing participation.

**Table 1 ijerph-17-04562-t001:** Participants’ demographics.

		Number	Percentage (%)
Gender	Male	7	44
	Female	9	56
Carer relationship	Spouse	6	75
	Child	2	25
Diagnosis	Alzheimer’s Disease	5	62
	Vascular Dementia	1	12
	Parkinson’s Dementia	1	12
	Lewy Body Dementia	1	12
Ethnicity	White-British	16	100

**Table 2 ijerph-17-04562-t002:** Outcome measure statistics at baseline and follow up.

Outcome Measure	Baseline (*n* = 16) (mean ± SD)	Follow Up (*n* = 16) (mean ± SD)	Mean Difference [95% CI]
**Cognition**			
HVLT Recall (/36) ^a^	15.63 ± 10.85	16.75 ± 11.48	1.13 [−2.74, 0.22]
HVLT Recognition (/12) ^a^	7.88 ± 4.94	7.94 ± 3.80	0.06 [−1.80, 1.67]
**Quality of Life**			
DEMQoL (/116) ^a^	93.00 ± 10.33	92.00 ± 11.60	−1.00 [−4.62, 6.62]
**Mobility**			
TUG (secs) ^b^	9.72 ± 3.67	11.02 ± 8.44	−1.29 [−5.28, 2.69]
Berg Balance (/56) ^a^	50.38 ± 6.29	49.13 ± 6.90	−1.25 [−0.43, 2.93]
**Strength**			
Grip Strength R (kg) ^a^	24.44 ± 6.76	24.15 ± 6.19	−0.29 [−1.53, 2.11]
Grip Strength L (kg) ^a^	22.81 ± 5.48	23.69 ± 7.28	0.89 [−2.98, 1.21]
**Independence**			
NEADL (/22) ^a^	14.31 ± 7.51	13.56 ± 8.30	−0.75 [−06–1.56]
**Mood**			
HADS (/42) ^b^	13.00 ± 6.69	11.67 ± 7.90	1.33 [−1.44, 4.11]
**Physical Activity**			
LAPAQ (/MET hrs/wk) ^a^	45.42 ± 28.53	49.86 ± 41.03	4.44 [−23.03, 14.14]
**Loneliness**			
UCLA Loneliness Scale (/80) ^b^	44.17 ± 10.45	42.42 ± 10.79	1.75 [−1.24, 4.74]

a: higher scores better, b: lower scores better. Notes: Hopkins Verbal Learning Test (HVLT); Dementia Quality of Life Scale (DEMQoL); Timed up and Go (TUG); Nottingham Extended ADL Scale (NEADL); Hospital Anxiety and Depression Scale (HADS); LASA Physical Activity Questionnaire (LAPAQ).

**Table 3 ijerph-17-04562-t003:** Participants Demographics.

Pseudonym	Carer/PWD	Diagnosis	Gender
FG1 P1	Carer	*n*/a	Female
FG1 P2	PWD	Vascular Dementia	Female
FG1 P3	Carer	*n*/a	Female
FG1 P4	PWD	Alzheimer’s	Male
FG2 P1	Carer	*n*/a	Female
FG2 P2	PWD	Alzheimer’s	Male
FG2 P3	Carer	*n*/a	Female
FG2 P4	PWD	Alzheimer’s	Male
FG2 P5	Carer	*n*/a	Male
FG2 P6	PWD	Alzheimer’s	Female

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
