# Peer review of "The Impact of a Dementia-Friendly Exercise Class on People Living with Dementia: A Mixed-Methods Study"

_ijerph, 2020, doi:10.3390/ijerph17124562_

Round 1
Reviewer 1 Report
I found the research interesting and well presented. The use of mixed methods data and the full explanation of findings was important and useful.
My recommendation is that the justification of the exercise methods could be extended. There was clearly no attempt at a co-design of the class or perhaps choice of exercises and so it would be important to identify why core strength and resistance work was justified and then linked to your choice of outcome metrics. This is an exercise class but it is unclear why it is an adapted environment? Is it in the level of staffing or explanation of exercises? what makes it adapted?
The introduction to the group class and the recruitment methods for participants is also less well explained. I assume that factors like charging and membership fees, sometimes barrier to participation were not a consideration. It was unclear whether this was an NHS treatment intervention- implying a referral process or a class offered by the University. Please state and identify if this was purposive sampling within the larger group.
The discussion is reasonable but it might be possible to add detail about the beneficial effects of environment and support from volunteers- some further description could perhaps be offered to identify the specific reason for this being dementia-friendly. Most sport classes which are accessible offer additional environmental features to adapt to particular age need, there is perhaps a need to use the data and further literature to suggest characteristics or capabilities that are particular to needs of people with dementia
Author Response
Responses to Reviewer 1
Many thanks for taking the time to read and review our article. We have considered your comments and have provided the following responses.
Comment 1: My recommendation is that the justification of the exercise methods could be extended. There was clearly no attempt at a co-design of the class or perhaps choice of exercises and so it would be important to identify why core strength and resistance work was justified and then linked to your choice of outcome metrics. This is an exercise class but it is unclear why it is an adapted environment? Is it in the level of staffing or explanation of exercises? what makes it adapted?
Response 1: Thank you for highlighting the need for additional information on the existing exercise class. Additional information has been included to demonstrate the rationale for the content of the class and its adaptations for people with dementia (pg 2/3 Setting).
Comment 2: The introduction to the group class and the recruitment methods for participants is also less well explained. I assume that factors like charging and membership fees, sometimes barrier to participation were not a consideration. It was unclear whether this was an NHS treatment intervention- implying a referral process or a class offered by the University. Please state and identify if this was purposive sampling within the larger group.
Response 2: We have added information regarding charging and (lack of) referral needed (Pg 2 Lines 80-83). The lead author was introduced to the class during a session and the study explained. Everyone who attended the class who could provide informed capacity was invited to participate and information sheets left with those who were interested. There was no purposive sampling - everyone was invited to participate and those expressed an interest provided an information sheet and everyone who consented took part.
Comment 3: The discussion is reasonable, but it might be possible to add detail about the beneficial effects of environment and support from volunteers- some further description could perhaps be offered to identify the specific reason for this being dementia-friendly. Most sport classes which are accessible offer additional environmental features to adapt to particular age need, there is perhaps a need to use the data and further literature to suggest characteristics or capabilities that are particular to needs of people with dementia.
Response 2: Additional references and considerations have been added to the discussion as suggested to highlight the alignment of the class with dementia friendly communities and existing literature on the role of the volunteer (pg 11, lines 333, 340,363).
Reviewer 2 Report
This is a very relevant manuscript for the field of physical activity and exercise for people with dementia. More dementia-friendly exercise classes are needed around the world; and this work adds valuable information to make it possible.
There are few points that need to be addressed to improve this work.
Abstract
- The abstract is very clear, however some data (numbers) would improve it, namely the positive mean differences of the physiological and cognitive outcome measures.
Introduction
- Some information about the low physical activity levels and/or the low adherence to exercise classes of people with dementia would enrich your argument for the need of friendly exercise interventions. I would suggest linking the first and the second sentences with that information.
- Line 44: This is a very good increase in attendance. Did it happen because family carers where included? Or because this is a tailored programme? Does the programme have other characteristics that might improve the attendance?
- Line 52: muscle strength
- Line 53: well-being is not mentioned on methods nor results.
Materials and methods
- It is not clear if people without a diagnosis of dementia were included in the study. Where defined exclusion criteria?
- Line 74 …the class meeting with those criteria...
- Information about the duration of the programme and the duration of each exercise class is missing. The role of the carers during the exercise classes is also unclear.
- Line 89: “the same exercise each week and can be completed at different levels, depending on individual physical capability” and depending on individual progress along the programme?
- It seems that the participants were already taking part of the exercise classes. Is it right? For how long? If they started at different time points, can it influence the results?
- Line 108: Muscle strength
- Were the measures repeated three-months later performed by the same researcher?
- Please add references to justify the methods used in the focus group (i.e., interview with flexible approach, notes, transcription, and audio record)
Data analysis
- The results of this manuscript would be enriched with inferential statistics (e.g., dependent t-test or Wilcoxon Signed Rank Test). If a statistical test was already conducted please add the information in data analysis section.
Results
- Please provide information about the age of the participants (people with dementia and their carers). Table 1 and text.
- Lines 162 & 163: can you further explain this sentence, please? which measures where not complete and reasons (if available)? If were questionnaires, is there possible that it could be avoid if the dyad completed them with the researcher?
- Line 170: which statistical test was used? this information is missing in the data analysis section.
- Table 2: TUG has a -1.29 s mean difference. Is it right? Or should be 1.29? The same for mood and loneliness.
- Table 2: please add the legend of the abbreviations at the end of the table 2.
- Lines 179 & 180: negative signs for loneliness and mood (-1.75 and -1.33)
Discussion
- Lines 345-348 this is indeed a limitation and some measures did not change because people already achieved their improvements and are maintaining their benefits now. More information about when the participants were integrated in the exercise classes would help to understand the results.
- Lines 350-352 A control group should be considered in future studies.
- Figure 1: some words are missing in the description of the exercise class characteristics.
Author Response
Responses to Reviewer 2
Many thanks for taking the time to read and review our article. We have considered your comments and have provided the following responses.
Comment 4: The abstract is very clear, however some data (numbers) would improve it, namely the positive mean differences of the physiological and cognitive outcome measures.
Response 4: Thank you for your suggestion. The positive mean differences for the outcomes mentioned in the abstract have now been added.
Comment 5: Some information about the low physical activity levels and/or the low adherence to exercise classes of people with dementia would enrich your argument for the need of friendly exercise interventions. I would suggest linking the first and the second sentences with that information.
Response 5: Thank you for this suggestion and a reference to a systematic review on adherence to group exercises in people living with dementia has been added to the manuscript (pg 1 lines 36-39).
Comment 6: Line 44: This is a very good increase in attendance. Did it happen because family carers where included? Or because this is a tailored programme? Does the programme have other characteristics that might improve the attendance?
Response 6: Thank you for your comments. New people tended to attend the class after hearing about it through word of mouth. The reasons for continued attendance was discussed in the focus groups and has been explored in lines 248, 251, 303, 304, 324-326.
Comment 7: Line 52: muscle strength
Response 7: This has been amended
Comment 8: Line 53: well-being is not mentioned on methods nor results
Response 8: Thank you for bringing this to our attention. Wellbeing has now been removed as we did not complete a specific wellbeing outcome measure
Comment 9: It is not clear if people without a diagnosis of dementia were included in the study. Where defined exclusion criteria?
Response 9: Section 2.2 has been rephrased to make the inclusion criteria clearer.
Comment 10: Line 74 …the class meeting with those criteria...
Response 10: This line has also been rephrased for clarity.
Comment 11: Information about the duration of the programme and the duration of each exercise class is missing. The role of the carers during the exercise classes is also unclear.
Response 11: The exercise class is a rolling programme where attendees are not limited to the number of sessions or length of time that they can attend. Thank you for raising this comment and additional information has been added into the manuscript to make this clear (pg 2 lines 80 – 83)
Comment 12: Line 89: “the same exercise each week and can be completed at different levels, depending on individual physical capability” and depending on individual progress along the programme?
Response 12: Thank you for bringing this point to our attention. It is indeed the case the participants can progress through to different levels as they improve, and this has been added to the manuscript to make this clear (pg 3 line 92)
Comment 13: It seems that the participants were already taking part of the exercise classes. Is it right? For how long? If they started at different time points, can it influence the results?
Response 13: Thank you for your comments. Yes, the participants were already attending the exercise classes. The classes started in September 2017 and at least one dyad in the study had been attending since then. We also had at least one other dyad who had only started attending a couple of weeks prior to the study starting. The time that they have been at the class would indeed have influenced the results as you would assume that people who had been attending longer may have already reached a plateaux whilst those who had recently started may have improved more. This has now been included as a study limitation in the discussion (pg 11 line 360)
Comment 14: Line 108: Muscle strength
Response 14: This has been amended
Comment 15: Were the measures repeated three-months later performed by the same researcher?
Response 15: Thank you for bringing this to our attention. Yes, the measure were completed by the same researcher and additional information has been added to the manuscript to reflect this (pg 3 line 126)
Comment 16: Please add references to justify the methods used in the focus group (i.e., interview with flexible approach, notes, transcription, and audio record)
Response 16: Thank you for your comments. References have been added (pg 3 Line 133)
Comment 17: The results of this manuscript would be enriched with inferential statistics (e.g., dependent t-test or Wilcoxon Signed Rank Test). If a statistical test was already conducted please add the information in data analysis section.
Response 17: Thank you for your suggestion. However, due to the small sample number we completed descriptive statistics only and additional information has been added to the manuscript to make this clear (pg 5 line 180)
Comment 18: Please provide information about the age of the participants (people with dementia and their carers). Table 1 and text.
Response 18: Thank you for the comment. Date of birth was not recorded and therefore has not been included in the results. This is a limitation of the study and has been included within the discussion (pg 12 line 370)
Comment 19: Lines 162 & 163: can you further explain this sentence, please? which measures where not complete and reasons (if available)? If were questionnaires, is there possible that it could be avoid if the dyad completed them with the researcher?
Response 19: Thank you for your suggestion. This information has been added to the manuscript (pg 4 lines 169 – 172)
Comment 20: Line 170: which statistical test was used? this information is missing in the data analysis section.
Response 20: Thank you for bringing this to our attention. As inferential statistics were not used the manuscript has been amended to remove the line about significance and replaced with a line that only descriptive statistics were completed due to the small sample size (pg 5 line 180)
Comment 21: Table 2: TUG has a -1.29 s mean difference. Is it right? Or should be 1.29? The same for mood and loneliness.
Response 21: Thank you for your comments. Yes, the TUG is correct as it worsened over the 3-month period (it took longer to do it) so it is a negative figure. Mood and loneliness improved and as indicated by the table these scores get lower as they improve so are positive numbers.
Comment 22 Table 2: please add the legend of the abbreviations at the end of the table 2.
Response 22: Thank you for your suggestion. The abbreviations have now been added to the legend for Table 2.
Comment 23: Lines 179 & 180: negative signs for loneliness and mood (-1.75 and -1.33)
Response 23: Thank you for your comments. The scores for mood and loneliness decrease as they get better (see legend Table 2 b lower scores better) so these results indicate a positive change.
Comment 24: Lines 345-348 this is indeed a limitation and some measures did not change because people already achieved their improvements and are maintaining their benefits now. More information about when the participants were integrated in the exercise classes would help to understand the results.
Response 24: Thank you for your suggestion. Additional information has been added to the manuscript to reflect the differing lengths of attendance to the limitations of the study (pg 11 lines 360 – 363)
Comment 25: Lines 350-352 A control group should be considered in future studies.
Response 25: Thank you for your comment. Although this was an exploratory study a line has been added to the manuscript on future research including control groups (pg 12 line 373)
Comment 26: Figure 1: some words are missing in the description of the exercise class characteristics.
Response 26. Thank you for your comment. I have checked the manuscript and I can’t see anything that is missing in Figure 1. Not all the exercises have explanations regarding increasing the difficulties.
Reviewer 3 Report
The study represents an area of increasing need.
Could the conclusions be tightened up to provide a take home message. For example key features being dementia specific, environment but really about the parking, facilities rather than being a sports centre.
This could be in the discussion so providing key elements for success, as the discussion is rather long.
Author Response
Responses to Reviewer 3
Many thanks for taking the time to read and review our article. We have considered your comments and have provided the following responses.
Comment 27: Could the conclusions be tightened up to provide a take home message. For example key features being dementia specific, environment but really about the parking, facilities rather than being a sports centre. This could be in the discussion so providing key elements for success, as the discussion is rather long.
Response 27: Thank for your suggestion. We have added additional information to the conclusion to specify the key characteristics of the classes to provide a take home message.